# Development of a Fast Convergence Gray-Level Co-Occurrence Matrix for Sea Surface Wind Direction Extraction from Marine Radar Images

Hui Wang [1], Shiyu Li [2,*], Haiyang Qiu [1], Zhizhong Lu [3], Yanbo Wei [4], Zhiyu Zhu [2] and Huilin Ge [2]

1   School of Naval Architecture and Ocean Engineering, Guangzhou Maritime University, Guangzhou 510725, China; heu_wanghui@126.com (H.W.)
2   School of Automation, Jiangsu University of Science and Technology, Zhenjiang 212003, China
3   College of Intelligent Systems Science and Engineering, Harbin Engineering University, Harbin 150001, China
4   College of Physical and Electronic Information, Luoyang Normal University, Luoyang 471022, China
*   Correspondence: lishiyu103@163.com

**Abstract:** The new sea surface wind direction from the X-band marine radar image is proposed in this study using a fast convergent gray-level co-occurrence matrix (FC-GLCM) algorithm. First, the radar image is sampled directly without the need for interpolation due to the algorithm's application of the GLCM to the polar co-ordinate system, which reduces the inaccuracy caused by image transformation. An additional process is then to merge the fast convergence method with the optimized GLCM so that the circular transition between rough and fine estimates is acquired, resulting in the fast convergence and accuracy improvement of the GLCM. Furthermore, the algorithm will affect the GLCM spatial distribution while calculating it, and it can automatically resolve the 180° ambiguity problem of sea surface wind direction retrieved from radar images. Finally, the proposed method is applied to 1436 X-band marine radar sequences collected from the coast of the East China Sea. Compared with in situ anemometer data, the correlation coefficient is as high as 0.9268, and the RMSE is 4.9867°. The new method was also tested under diverse sea conditions. The FC-GLCM wind direction results against the adaptive reduced method (ARM), energy spectrum method (ESM), and the traditional GLCM (T-GLCM) method produced the best stability and accuracy, in which the RMSE decreased by 91.6%, 67.7%, and 18.1%, respectively.

**Keywords:** fast convergent GLCM; wind direction; X-band marine radar image; 180° ambiguity problem

## 1. Introduction

Sea surface wind estimation plays an important role in many marine activities, such as navigation safety, performance, and marine exploration [1]. Accurate wind direction information is not only an important judgment factor for ship sailing direction, but also an important factor for military guidance and carrier-borne aircraft takeoff and landing safety. The conventional in situ wind sensors such as anemometers placed in ships and buoys are susceptible to airflow distortion caused by superstructures or the movement of the anemometer platform, and can provide inaccurate wind data [2]. Other remote sensing methods such as scatterometers [3,4] and radiometers [5,6] can achieve a wider coverage of wind information, but they cannot perform high-precision small-area measurements. Synthetic aperture radar (SAR) has high resolution and day-and-night imaging capabilities, but its computational cost is too high [7].

Land-based and ship-borne low-cost X-band marine radars with high temporal and spatial resolution have been broadly used in the observation of ocean wave height [8], currents [9], rain [10], and sea surface winds [11]. The average wind field information retrieved from X-band marine radar is one of the more important environmental factors

for continuous and real-time takeoff and landing safety of carrier-borne aircraft under the conditions of sea dynamic vehicles. The average wind information near 1 km from the ship is free of structural disturbance and tends to be closer to free flow. It provides most accurate steady-state wind parameters on the ship's surface for carrier aircraft [12,13].

Previous studies have demonstrated that, for X-band radar operating at grazing incidence with horizontal transmit–horizontal receive (HH) polarization, there are two methods used to retrieve wind direction information. The first method, proposed in 1998, provides that the polarization radar cross section (RCS) is at its maximum when the wind is upwind, and the sea surface wind direction information can be retrieved based on this principle [14]. In 2012, Lund et al. proposed that the harmonic function model could be established by applying the RCS of the marine radar image and sea surface wind field, and the unique maximum value of the function could be more accurately obtained [15]. In 2017, the relation between the inherent modal function component of the radar image and the sea surface wind field function was established based on the ensemble empirical mode decomposition method and the sea surface wind direction information was retrieved based on the unique maximum value of the mode function [16]. However, these methods cannot obtain an accurate upwind peak value for the radar data without 360° coverage. In 2020, Chen, X. and Huang, W. et al. [17,18] successively identified the relationship between radar images and the wind field in various deep learning-based models and directly obtained sea surface wind direction information from radar images according to these models. This type of method results in models with significant differences for different radar models, requiring readjustment of model parameters or modification of the model function.

The second method is based on the characteristics of small-scale wind streak in the marine radar image sequences to retrieve wind direction information. The static small-scale wind streak can be extracted from the nautical radar image sequence and the wind streak is aligned with the wind direction [19,20]. Two methods have been developed for texture orientation estimation, including the most commonly used spectrum-based and gradient-based methods. Several spectrum-based methods have also been developed, including those based on Fourier transform [21], Radon transform [22], the Gabor filter [23], and wavelet transform [24]. In 2016, Wang, Y. and Huang, W. [25] proposed that sea surface wind direction information can be obtained under rainfall and nonrainfall conditions using the Fourier transform wavenumber spectrum region and the value of the radar image, respectively. A wind field energy spectrum method (ESM) with the characteristic that the axis of the small-scale wind streak is parallel to the sea surface wind direction is proposed in [26]. However, this kind of method can only be used when the image coverage is greater than 180°, which is highly dependent on the streak scale characteristics. In special weather or when the streak scale is outside the estimated range, the accuracy is greatly reduced and some data are not applicable.

For gradient-based methods, the main local orientation of the wind streak is estimated by assuming that it is perpendicular to the wind direction. The smoothing image and calculating gradient are combined in the frequency domain to reduce the influence of noise [27]. A gradient optical flow method to retrieve sea surface wind field information based on gust signal characteristics in marine radar images is proposed in [28]. However, as wind field is a static characteristic signal, it is difficult to extract the spatial characteristics from a single image, resulting in the retrieval accuracy of the optical flow method being unable to meet the engineering requirements. A local gradient method (LGM) to retrieve sea surface wind direction information from sea surface static feature images based on the wind streak feature is proposed in [29]. Although the retrieval accuracy of this method meets the engineering requirements, there remains a high volume of inapplicable data. An adaptive reduction algorithm-improved local gradient algorithm (ARM) is proposed in [30]. These methods are still dependent on the scale characteristics of the wind streak and the latter two methods still rely on the first method to solve the 180° ambiguity problem. In addition, the FT- and gradient-based methods usually require a large rectangular dataset with many samples of the texture to accurately calculate the direction. However, the

nautical radar image involves polar co-ordinates, which need to be transformed into rectangular data through image interpolation. This transformation process will cause radar image distortion and lead to inaccurate results. In 2021, Wang, H. et al. [26] proposed the wavenumber energy spectrum method (ESM) to directly obtain the sea surface wind direction information and better accuracy and data applicability were achieved. These methods are only used when the image coverage is greater than 180° and the accuracy is greatly reduced in special weather or the streak scale is not in the estimated range.

The paper proposes GLCM-based wind streak analysis to overcome the problems associated with FT- and gradient-based methods. However, the conventional GLCM for texture orientation estimation has some limits, as follows [31–33]: ① Identification range limitation. The traditional gray-level co-occurrence matrix calculation method can only identify the angles of 90°, 45°, 0°, and −45° in the horizontal direction. However, only parts of the radar image textures are located in these particular directions in practice, resulting in the gray-level co-occurrence being typically insufficient to accurately resolve all texture directions. ② Limited application range. The traditional gray-level co-occurrence matrix calculation is only limited to Cartesian co-ordinate systems, but marine radar use rotating scanning for imaging, and the generated images are based on polar coordinates. ③ Insufficient directional accuracy. The confidence interval of the traditional gray-level co-occurrence matrix is 10°, while the radar image amplitude resolution is 0.1°. In this paper, a FC-GLCM based on the wind streak image method is proposed; the FC-GLCM angle estimation of the circular transition between rough and fine transition is performed for each set separately and adaptive angle fusion is performed according to the clustering results of the angle categories. The overall trend of the identified image is the main wind direction information. This new method not only meets the requirements of high accuracy of wind field information extraction in engineering, but also leads to the change in spatial distribution of the matrix when the normalized GLCM calculation is carried out. It can automatically solve the 180° ambiguity problem of retrieving sea wind direction information in the space domain.

The remainder of this paper is organized as follows. An overview of the data and polar co-ordinate sea surface static feature image extraction process is presented in Section 2. In Section 3, the new wind direction retrieval method, the FC-GLCM, is proposed and how to solve the 180° ambiguity problem in the retrieval of sea surface wind direction information is presented. In Section 4, the method is applied to real radar images to validate the wind direction results, and the sensitivity and robustness are discussed in comparison with the traditional methods. Finally, a summary is given in Section 5.

## 2. Data Overview

### 2.1. Data Source

A typical horizontal polarization short pulse mode X-band marine radar was employed in this paper. For the X-band radar, the main backscatter mechanism at the ocean surface is Bragg scattering when the NRCS (normalized radar cross section) is proportional to the spectral density of the surface roughness [19]. The X-band radar antenna has a compact bulk and excellent azimuth discrimination, making it ideal for short-range detection. It not only emits electromagnetic waves, but also receives echo waves to form a sea clutter image. The radar wavelength is 3 cm, the operating frequency is 9320~9500 Hz, the actual observation distance is up to 4500 m, the azimuth resolution is 0.1°, and the radial resolution is 7.5 m. It should be noted that the radar operates in a rotational scanning mode. For each 0.1° interval, the radar antenna broadcasts and receives a column of electromagnetic wave signals, which are stored per line. Assuming that the ship's head direction is the initial harness, the azimuth of the succeeding storage lines is increased by 0.1°.

The marine radar revolves for one circle to gather an entire image, which takes roughly 2.5 s and includes up to 3300 ± 120 harnesses, with each of the harnesses containing about 600 pixels points on the radial direction line. On the onboard software operating platform, the radar scan image is transformed into a polar image. A sequence of time

series consisting of 32 images lasting 80 s is obtained from the radar pulse value. Given the incidence angle of the radar antenna and the height above sea level, a considerable distance of wave inversion will result in local information distortion, while a short distance will weaken the algorithm's robustness. Consequently, the ideal distance of the wave inversion is determined to be $600 \sim 2100$ m. Because of the marine radar's $360°$ surround measuring approach, some data along the coast will be included in the results. The isolated bright spots or continuous bright regions that emerge in the marine radar images is caused by fixed targets, as shown in Figure 1a. Too many fixed targets will not only obscure certain radar image features but will also enhance the mean value of the radar echo strength throughout the whole study region, which provides excellent coverage for the static signal provided by the sea surface wind field [34]. As a result, the interference of fixed objects in the radar image must be removed. The amplitude direction of $106° \sim -69°$ is removed in this research, and the resulting image is presented in Figure 1b. The initial angle is defined in this paper as the ship's head direction $\theta_n = 93°$, according to the features of the measured data. Eventually, the study area is clockwise $\dot{U}_0(\theta_n) = [106°, 291°]$ with a radial distance of $\dot{U}_0(l_n) = [600 \text{ m}, 2100 \text{ m}]$.

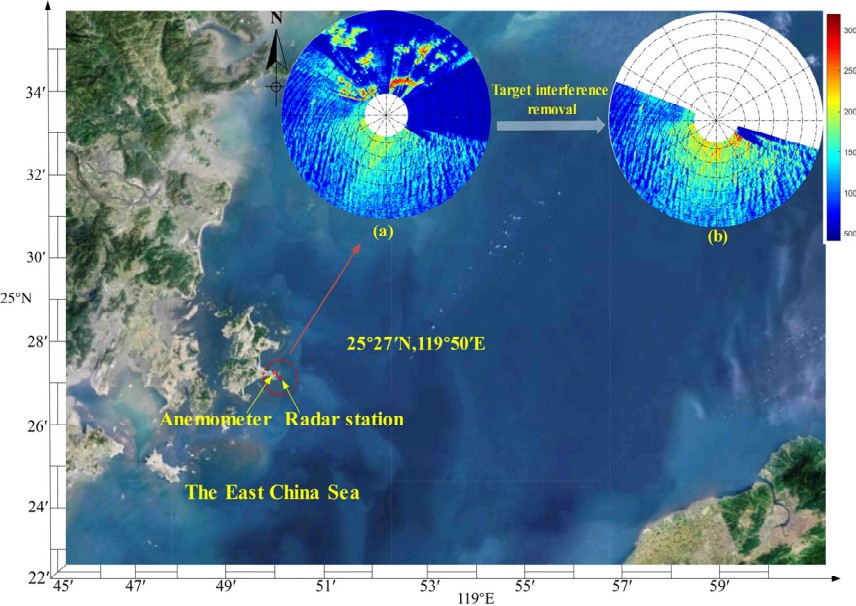

**Figure 1.** Map of the experiment site. Locations of the radar station and anemometer are shown by red stars. (**a**) Single polar marine radar image collected on 00:20 UTC+8 27 October 2010; (**b**) selected radar image after removing $106° \sim -69°$ amplitude on 27 October 2010.

The original radar images utilized in this paper were collected from the HEU wave and current monitoring systems located on Haitan Island along the East China Sea at $25°27'$N, $119°50'$E, between 22 October and 30 October 2010. Rainfall will increase the roughness of the sea surface and further surge the scattering area of the radio wave generated by the radar when it comes into contact with the surface, affecting the echo intensity of the sea surface and introducing noise to the radar images. After removing the data of strong rainfall by applying the wave texture difference method in [35], 1448 sets of marine radar sequences were selected from 1634 sets. After a thorough selection process, a total of 1436 sets of marine radar sequences were deemed suitable for analysis out of the 1448 sets. The exclusion of 12 sets was necessary due to the low wind speed (1–2 m/s) present in the data. As shown in Figure 2a, when the wind speed is normal (>2 m/s), the wind streaks in the image show a clear trend and the red part dominates the entire image. The red and blue parts have clear boundary lines, which can effectively extract dominant feature groups in subsequent image preprocessing. When the wind speed is too low, as shown in Figure 2b, almost all parts of the entire image are dark blue and the trend of wind

streaks is not obvious. Only the innermost ring has a slight light blue part, which leads to the inability to extract feature groups in subsequent image preprocessing processes. In this case, the method in this paper cannot be applied to retrieve the wind direction. An in situ Model-05103 wind field monitor was used to gather reference wind directions for validating the proposed scheme for wind direction information extraction from the marine radar. Since the Model-05103 wind field monitor records every minute, the wind direction at the moment of the image sequence is selected as the reference wind direction for the research.

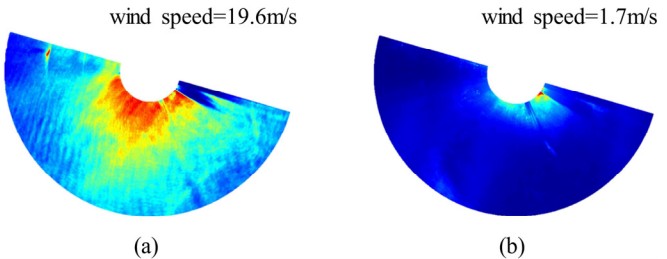

(a)          (b)

**Figure 2.** Static feature images of sea surface under different wind speeds, different color areas as the bar of Fig.4, the obtaining process is in the Section 2.2.2. (**a**) The wind speed during measurement was 19.6 m/s; (**b**) the wind speed during measurement was 1.7 m/s.

### 2.2. Polar Co-Ordinate Sea Surface Static Feature Image

#### 2.2.1. Co-Ordinate System Transformation

Since polar co-ordinate systems and image polar co-ordinate systems have different definitions of central co-ordinates and co-ordinate axis directions, co-ordinate system transformation is required before image preprocessing; the transformation process is shown in Figure 3. As shown in Figure 3a, the origin of the polar co-ordinate system is located in the image center, the north direction is the positive direction of the polar axis, and the square of the offset angle is clockwise. The co-ordinates of the point are $(\rho, \theta)$, the distance from the point to the origin is $\rho$, and the red dotted line angle relative to the north is $\theta$. For the same point, the absolute position is the same under different co-ordinate systems, but the relative position is different. The origin of the image polar co-ordinate system is located at the upper left corner of the image, as shown in Figure 3b, and the horizontal direction is the positive direction of the polar axis. The co-ordinates of this point are $(\rho', \theta')$ as shown in Figure 2b, the distance from the point to the origin is $\rho'$, and the red dotted line relative to the horizontal axis is $\theta'$. The co-ordinates in the original polar co-ordinate system can be converted to the co-ordinates in the image polar co-ordinate system using the following formula:

$$\begin{cases} \rho' = \sqrt{2R^2 + \rho^2 + 2\rho R(\sin\theta - \cos\theta)} \\ \theta' = \arcsin\dfrac{R - \rho\cos\theta}{\sqrt{2R^2 + \rho^2 + 2\rho R(\sin\theta - \cos\theta)}} \end{cases} \tag{1}$$

where $(\rho, \theta)$ is the co-ordinate pair of the point in polar co-ordinates, $(\rho', \theta')$ is the co-ordinate pair of the point in image polar co-ordinates, and $R$ is the radius of the image area. The radius of the area selected in this paper is $R = 2100$ m.

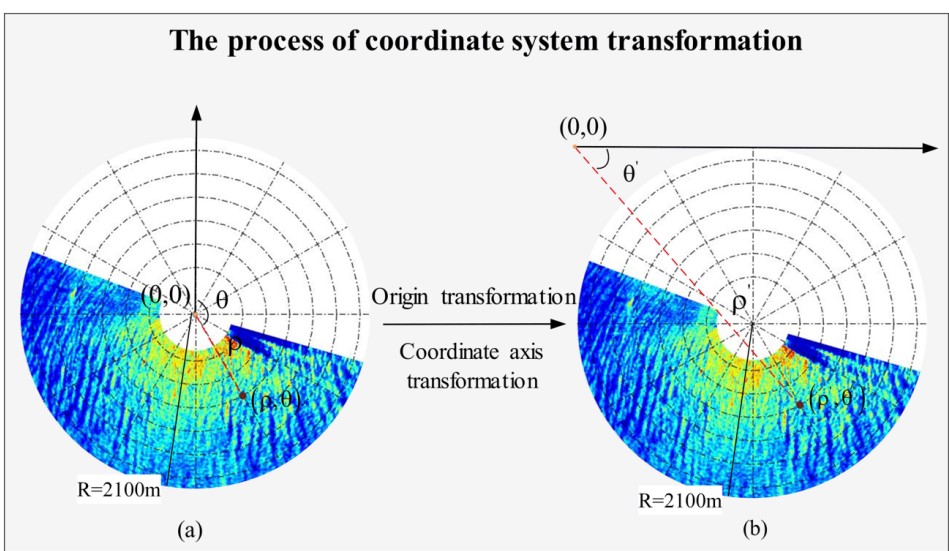

(a)

(b)

**Figure 3.** Process of co-ordinate system transformation on 00:20 UTC+8 27 October 2010. (**a**) Polar co-ordinate system; (**b**) image polar co-ordinate system.

### 2.2.2. Process of Extracting Sea Surface Static Features

According to the imaging mechanism of the X-band marine radar, the modulation of the sea surface wind field leads to the feature of small-scale wind streaks in the radar image sequence. The small-scale wind streaks have static features that make them hard to detect on a signal radar image since the signal observation period of the marine radar is only about 2.5 s. As a consequence, in order to obtain the sea surface static features including the sea surface wind field, it is essential to overlay and average the radar image sequence in each period ($\approx$80 s). The static patterns of the sea surface features with low-frequency signals (including sea surface wind field information) were obtained after filtering out high-frequency wave noise. The main wind direction of the selected area is obtained based on the characteristic that the small-scale wind streak is aligned with the sea surface wind direction [19]. In this work, 32 image sequences are overlain and averaged to form a sea surface static feature image, the process of which is shown in Figure 4. In this paper, the radar data digitized and stored the spatial and temporal radar backscatter information as a sequence of images with a 14-bit grayscale depth, i.e., digitized backscatter intensities ranging from 0 to 8192. In order to perform the following steps, the sea surface static feature image rescaled to the gray scale [0, 255] (8-bit) $\mathcal{I}$ as in that image.

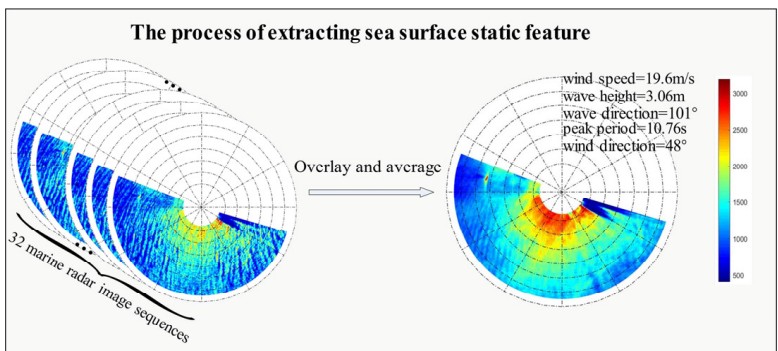

**Figure 4.** Process of overlaying and averaging 32 radar image sequences collected on 00:20 UTC+8 27 October 2010, obtaining a sea surface static feature image (including sea surface wind field information). The wind speed during measurement was 19.6 m/s, the wave height was 3.06 m, the wave direction was 101°, the peak period was 10.76 s, and the wind direction was 48°.

## 3. Sea Surface Wind Direction Extraction Algorithm

This section primarily describes the framework of the proposed FC-GLCM algorithm, as shown in Figure 5. The algorithm's procedure is divided into three stages: image preprocessing of polar co-ordinate sea surface static features, the calculation of a fast-convergence gray-level co-occurrence matrix, and adaptive trend fusion.

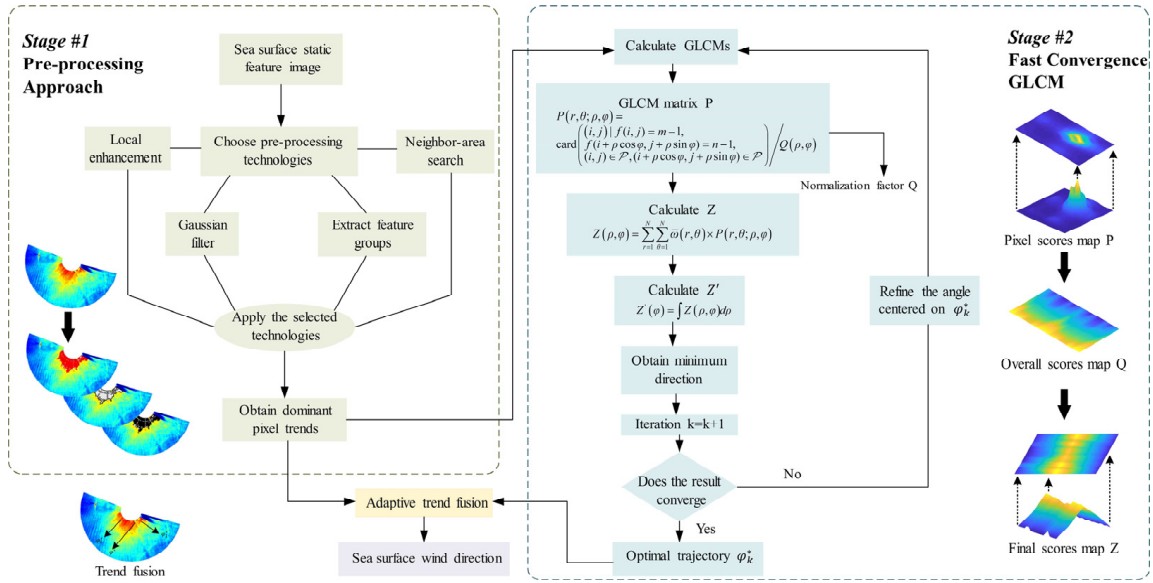

**Figure 5.** The framework of the proposed FC-GLCM algorithm.

### 3.1. Image Preprocessing of Polar Co-Ordinate Sea Surface Static Features

The pixel value distribution is clustered, and the prevalent pixel categories are identified based on the features of the sea surface static feature image. The prevalent pixel categories extracted are enhanced [36]. When a sea surface static feature image turns to RGB, $\mathcal{I} \in \mathbb{R}^{\omega \times h \times 3}$ is input and the pixel-level cumulative illumination value $\mathcal{L}^c(r, \theta, \mathcal{I})$ can be formulated as:

$$\mathcal{L}^c(r, \theta, \mathcal{I}) = \sum_m \alpha_m \Psi_m(\mathcal{I}(r, \theta)), m \in \{R, G, B\} \tag{2}$$

where $\Psi_m(\mathcal{I}(r, \theta))$ denotes the pixel value of image $\mathcal{I}$ at location $(r, \theta)$ in channel m and the channel wise weight parameters $\alpha_R, \alpha_G$, and $\alpha_B$ is the proportion of the corresponding $R, G, B$ pixels in channel to the total pixels of the image, and jointly meet $\alpha_R + \alpha_G + \alpha_B = 1$. By employing different weights on the $R, G, B$ channels, high-contrast-rate colors such as yellow and orange will be suppressed and low-contrast-rate colors such as red will be amplified in the image $\mathcal{I}$ as $\mathcal{L}^c(r, \theta, \mathcal{I})$. Then, the log-average cumulative luminance $\widetilde{\mathcal{L}}^c(\mathcal{I})$ is given as in [37]:

$$\widetilde{\mathcal{L}}^c(\mathcal{I}) = \frac{1}{N_{\mathcal{I}}} \exp\left(\sum_{r, \theta} \log(\delta + \mathcal{L}^c(r, \theta, \mathcal{I}))\right) \tag{3}$$

where $N_{\mathcal{I}}$ is the total number of pixels in the image and $\delta$ is a relatively small quantity to avoid a zero value of $\mathcal{L}^c(r, \theta, \mathcal{I})$ in $\log(\delta + \mathcal{L}^c(r, \theta, \mathcal{I}))$. Eventually, the adaptive enhancement factor map $\mathcal{L}_g(r, \theta, \mathcal{I})$ can be obtained as:

$$\mathcal{L}_g(r, \theta, \mathcal{I}) = \frac{\log\left(\mathcal{L}^c(r, \theta, \mathcal{I}) / \widetilde{\mathcal{L}}^c(\mathcal{I}) + 1\right)}{\log\left(\mathcal{L}^c_{\max}(\mathcal{I}) / \widetilde{\mathcal{L}}^c(\mathcal{I}) + 1\right)} \tag{4}$$

where $\mathcal{L}^c_{\max}(\mathcal{I}) = \max(\mathcal{L}^c(r,\theta,\mathcal{I}))$ denotes the maximum value of $\mathcal{L}^c(r,\theta,\mathcal{I})$. The aim of factor calculation is to further adaptively alter the local color value of three intensity channels at each pixel to realize image enhancement, as follows:

$$\Psi_{\mathrm{m}}(\mathcal{I}_{\mathrm{e}}(r,\theta)) = \Psi_{\mathrm{m}}(\mathcal{I}(r,\theta)) * \frac{\mathcal{L}_g(r,\theta,\mathcal{I})}{\mathcal{L}^c(r,\theta,\mathcal{I})}, \mathrm{m} \in \{R,G,B\} \tag{5}$$

where $\mathcal{I}_{\mathrm{e}}$ denotes the enhanced version of the original $\mathcal{I}$. The area in red that has been enhanced accurately highlights the features of wind field modulation. Through the derivation operator, the local auto-enhancement image $\Theta_{\mathcal{L}}(\mathcal{I})$ can be constructed as:

$$\Theta_{\mathcal{L}}(\mathcal{I}) = \mathcal{L}^c(\mathcal{I}) - \mathcal{L}^c(\mathcal{I}_{\mathrm{e}}) = \frac{\mathcal{L}^c(r,\theta,\mathcal{I}) - \log\left(\frac{\mathcal{L}^c(r,\theta,\mathcal{I})}{\widetilde{\mathcal{L}}^c(\mathcal{I})+1}\right)}{\log\left(\mathcal{L}^c_{\max}(\mathcal{I})/\widetilde{\mathcal{L}}^c(\mathcal{I})+1\right)} \tag{6}$$

According to $\mathcal{L}^c(r,\theta,\mathcal{I}) \in [0,1]$, the value of local auto-enhancement image $\Theta_{\mathcal{L}}(\mathcal{I})$ can thus vary beside the original image $\mathcal{L}^c(r,\theta,\mathcal{I})$. Thereby, the class of low contrast rate can be indicated as the informative local region. To be specific, the average value $\mu$ and standard deviation $\sigma$ of the low contrast rate in $\Theta_{\mathcal{L}}(\mathcal{I})$ are computed. Following a three-sigma criterion in statistics, pixels in the range $\mu \pm 3\sigma$ are considered the informative local region, while others are less informative and should be cast away after the preprocessing step for higher efficiency.

To acquire the pixel where the dominant trend is situated in a locally enhanced marine radar image $\mathcal{I}_{\mathrm{e}}$, Gaussian differential filtering is applied to the enhancement image [37]. After Gaussian differential filtering, the image's two-dimensional Gaussian spectral function $G(\mathcal{L}_g(r,\theta))$ satisfies the following formula [38]:

$$G(\mathcal{L}_g(r,\theta)) = G(\mathcal{L}_g(r,\theta))_0 - G(\mathcal{L}_g(r,\theta))_{k_1} = \frac{1}{2\pi\sigma_0^2}e^{-\frac{(i^2+j^2)}{2\sigma_0^2}} - \frac{1}{2\pi\sigma_{k_1}^2}e^{-\frac{(i^2+j^2)}{2\sigma_{k_1}^2}} \tag{7}$$

where $\mathcal{L}_g$ is the Gaussian differential filtering of the locally enhanced image $\mathcal{I}_{\mathrm{e}}$ and the gray portion of the Gaussian filter can smooth the image and reduce noise. $G(\mathcal{L}_g(r,\theta))_0$ is the two-dimensional Gaussian spectral function under the initial phase angle. $G(\mathcal{L}_g(r,\theta))_k$ is the two-dimensional Gaussian spectral function with a phase angle of $k_1$, which takes the $\sigma_k = k_1\sigma_0$, $\sigma_0 = 1.6$, and $k_1 = 0.8$.

The range is optimized in depth after the Gaussian differential filtered image $\mathcal{L}_g$ is searched in four neighborhoods. To produce the preprocessed marine radar image $\mathcal{L}_f$, two sets of feature groups are extracted. $\mathcal{L}_f$ is the final preprocessed radar image; the portion that is black after depth optimization can clearly and precisely extract wind field stripe information.

### 3.2. Fast-Convergence Gray-Level Co-Occurrence Matrix

The traditional GLCM only focuses on the scenarios of direction extraction under the Cartesian co-ordinate system, which is a malfunction for the polar co-ordinated problems such as wind direction extraction for radar images. On the other hand, the accuracy of the retrieved wind direction information based on marine radar images is restricted due to practical project demands. This paper proposes a FC-GLCM to address the endogenous problem of the GLCM in order to extend the applicability of traditional GLCM to polar coordinate systems and improve its convergence competence to meet practical requirements. The visualization of the algorithm is shown in Figure 5. The overall mathematical model of the FC-GLCM can be formulated as follows:

$$Z_{k+1} = \sup\left\{\frac{1}{Z} \middle| Z_k = \min_{Z,k} P_Z^k(r,\theta;\rho,\varphi)\right\}, 0 \le r \le r_h, \theta_s \le \theta \le \theta_e, 0 \le \rho \le \rho_h, \varphi_s \le \varphi \le \varphi_h \tag{8}$$

where $Z_k$ is the optimal value of the matrix $Z'$ in the FC-GLCM at $k$-th iteration, $\min_{Z,k} P_Z^k(r, \theta; \rho, \varphi)$ is the cost function for jointly considering the variable pair $(Z, k)$ to obtain the minimal $Z_k$, while $P_Z^k(r, \theta; \rho, \varphi)$ is the GCLM process for $r, \theta; \rho, \varphi$ applied to the $k$-th iteration and $Z_k$ in the FC-GLCM. $\sup\{\cdot\}$ is to obtain the supremum of the specific set. According to the above definitions, the value of $Z_k$ can also represent the optimization degree of $\varphi_k$; thus, the supremum $\sup\left\{\frac{1}{Z} \middle| Z_k = \min_{Z,k} P_Z^k(r, \theta; \rho, \varphi)\right\}$ denotes the minimal $Z$ among all $k$ iterations and can obtain the corresponding optimal $\varphi_k$. Furthermore, $r_h$ is the threshold for amplitude $r$, while $\theta_s$ and $\theta_e$ are the lower bound and upper bound for the argument $\theta$, respectively. It should be noted that the world polar co-ordinates pair $(r, \theta)$ can specify a co-ordinate in the polar image $G\left(\mathcal{L}_f\right)$. Similar variable setting rules are employed for the relative polar co-ordinate pair $(\rho, \varphi)$ and the corresponding $\rho_h, \varphi_s, \varphi_h$. Finally, the relative polar coordinate pair $(\rho, \varphi)$ is oriented at the coordinate designated by the global polar coordinate pair $(r, \theta)$.

To be specific, the GLCM under the polar co-ordinates system can be formulated according to the following statements.

The gray-level ranging section of an image $f(r, \theta)$ at position $(r, \theta)$ varies from 0 to $N - 1$ and the GLCM is a matrix of size $N \times N$. In the traditional GLCM direction estimation, only the GLCM matrix of a relative position is used, which corresponds to an individual pixel in the image. For a GLCM of the relative position $(\rho, \varphi)$ oriented at position $(r, \theta)$, its matrix element at $(r, \theta)$ can be calculated by counting the pixel pairs, as follows [39]:

$$P(r, \theta; \rho, \varphi) = \mathrm{card}\left( \begin{array}{l} (i, j) | f(i, j) = m - 1, \\ f(i + \rho \cos \varphi, j + \rho \sin \varphi) = n - 1, \\ (i, j) \in \mathcal{P}, (i + \rho \cos \varphi, j + \rho \sin \varphi) \in \mathcal{P} \end{array} \right) / Q(\rho, \varphi) \quad (9)$$

where $\mathrm{card}(\cdot)$ denotes the counting function whose output is the number of elements in a set and $(\rho, \varphi)$ represents a line orientated at the position $(r, \theta)$ with a polar amplitude value $\rho$ and an angle value $\varphi$ with respect to the horizontal orientation of the position $(r, \theta)$. Furthermore, $\mathcal{P}$ is the pixel set of the image and $f(r, \theta)$ is the gray level at position $(r, \theta)$ in the polar co-ordinate system. The normalization factor $Q(\rho, \varphi)$ is:

$$Q(\rho, \varphi) = \mathrm{card}\{(i, j) | (i, j) \in \mathcal{P}, (i + \rho \cos \varphi, j + \rho \sin \varphi) \in \mathcal{P}\} \quad (10)$$

where $Q(\rho, \varphi)$ is the number of pixel pairs that satisfy the relative position $(\rho, \varphi)$. Then, the gray-level co-occurrence matrix satisfies the following formula:

$$Z(\rho, \varphi) = \sum_{r=1}^{N} \sum_{\theta=1}^{N} \overline{\omega}(r, \theta) \times P(r, \theta; \rho, \varphi) \quad (11)$$

where $\overline{\omega}(r, \theta)$ is an $(r, \theta)$-oriented increasing function, which can be written as:

$$\overline{\omega}(r, \theta) = 1 + \exp(-d[(r, \theta) | (r_0, \theta_0)]) \quad (12)$$

where $d[(r, \theta) | (r_0, \theta_0)]$ is the distance measure function between $(r, \theta)$ and its orientation point $(r_0, \theta_0)$.

This paper uses the integral of $Z(\rho, \varphi)$ with respect to $\rho$, which can be calculated as:

$$Z'(\varphi) = \int Z(\rho, \varphi) d\rho \quad (13)$$

In general, the discrete form of $Z'(\varphi)$ can be formulated as:

$$Z'(\varphi) = \sum_{\rho=1}^{R} \sum_{r=1}^{N} \sum_{\theta=1}^{N} \overline{\omega}(r,\theta) \times P(r,\theta;\rho,\varphi) \tag{14}$$

where $R$ is the integration range for $\rho$.

The optimal orientation $\varphi_{\min}$, which denotes the minimum value of $Z'(\varphi)$, is calculated as the following formula:

$$\varphi_{\min} = \arg\left(\min_{\varphi} Z'(\varphi)\right) \tag{15}$$

As shown in Figure 6, the FC-GLCM is not meaningless to repeatedly calculate k times of GLCM but uses a method of coarse–fine estimation of cyclic iteration. During the first GLCM, the algorithm finds an optimal solution in the transparent semicircle region of the graph $\varphi_1^*$. At this time, the estimated wind direction is $47°$, which is located at the black line of the semicircle. The estimated wind direction for the first time above $\varphi_1^*$ is the center and the step size is half of the last time. The GLCM is calculated again and an optimal solution is found in the purple sector area in the figure to obtain the iterative updated estimated wind direction $\varphi_2^*$, which is $48°$. Finally, the estimated wind direction updated by the second iteration $\varphi_2^*$ is the center and the step size is half of the last time. The GLCM is calculated again and an optimal solution is found in the orange sector area of the figure. Finally, the estimated wind direction after three iterations $\varphi_3^*$ is $48.1°$. Experiments show that the algorithm converges to the unique optimal direction after three iterations. The mathematical proofs of the uniqueness of optimal direction and the convergence properties of the FC-GLCM are presented in Appendix A.

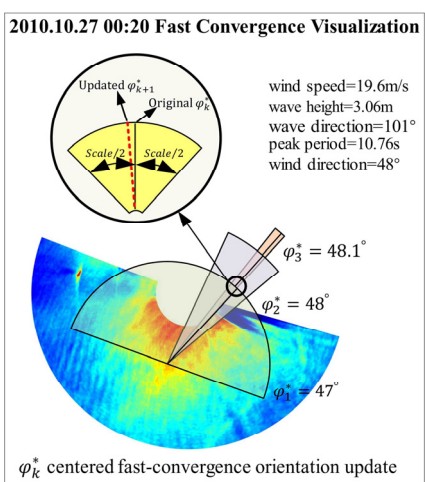

**Figure 6.** Fast-convergence visualization of radar image on 00:20 UTC+8 27 October 2010. (The sector part is the search area of each iteration and the circular part is the specific method of iterative update).

### 3.3. How to Improve the Efficiency of the FC-GLCM

According to Zheng et al. [33], the GLCM's wind direction estimation speed is distributed in $v_G = 0.05 - 0.1$fps, where fps is the frame rate, which represents the number of images the algorithm can process per second. The frame rate of the FC-GLCM algorithm proposed in this paper is distributed in $v_G = 0.03 - 0.07$fps through the actual computation of the experimental phase. In practical applications, the computation speed of the actual detection data of the direction station is distributed in $v_C = 0.2 - 0.35$fps. To improve the algorithm's efficiency and meet real-time requirements, this paper proposes two interpolation algorithms based on the features of the algorithm and the processed data.

According to the algorithm's properties, the following Table 1 compares the time complexity of the FC-GLCM with the traditional GLCM.

**Table 1.** Comparison of the temporal complexity of FC-GLCM with traditional GLCM.

| Step Title | FC-GLCM | Traditional GLCM |
|---|---|---|
| $P$ | $O(n^2)$ | $O(n^4)$ |
| $Q$ | $O(n^2 + \log n)$ | $O(n^2)$ |
| $Z$ | $O(n^2)$ | $O(n^4)$ |
| Preprocessing | $O(n^2)$ | - |

As shown in Table 1, the time complexity distribution of the FC-GLCM for each portion and the traditional GLCM is similar, with the primary amount of computation concentrated on the $P$ and $Z$ calculations. Due to the accuracy requirements of the algorithm, only the $P$ and $Z$ calculation processes with a large amount of calculation are interpolated. In terms of the interpolation implementation method, the sequence of interval calculation first and interpolation later is adopted. The specific implementation steps are as follows:

Step 1: Interval Calculation

According to the calculation formula of $P$, each variable's time complexity can be simplified to $O\left((n/c)^2\right)$, where $c$ is the interval order. Similarly, the $Z$ temporal complexity after the interval calculation is $O\left((n/c)^2\right)$.

In this paper, the mathematical expectation of the probability model distribution of the processed image is variable and there is no fixed distribution mode. Therefore, a robust and accurate interpolation approach must be required.

Step 2: Interpolation Completion

Approach 1: Kriging interpolation
All functions $P(x_1), P(x_2), \ldots, P(x_n)$ should have:

$$P^*(x_0) = f[P(x_1), P(x_2), \ldots, P(x_n)] \tag{16}$$

However, due to the complexity of the spatial variables analyzed by Kriging estimation technology changing with different spatial positions and the insufficient information given by the limited amount of observation data, determining the general form of the function clearly and totally is impossible, so we can only estimate $P^*(x_0)$, which is the form of $f$. When $f$ is considered as a linear function of $P(x_1), P(x_2), \ldots, P(x_n)$, the following results are obtained [40]:

$$P^*(x_0) = \lambda_0 + \sum_{i=1}^{n} P(x_i) \tag{17}$$

The limit of this estimation is based on the linear range. It is essential to decide which standard to apply for estimating the function before determining the constant $\lambda_0, \lambda_2, \ldots, \lambda_n$. The minimum variance is commonly employed as the estimate standard in a Kriging estimation scheme. The formula can be written as:

$$E\left\{[P(x_0) - P^*(x_0)]^2\right\} = \min \tag{18}$$

Kriging technology can be used for linear minimum variance estimation.

The pan-Kriging approach was chosen because the random function $P(x)$ involved in this study is neither defined nor constant.

Assume that the draft function $m(x)$ has the following form:

$$m(x) = \sum_{i=1}^{n} a_i f^i(x) \tag{19}$$

where $a_i$ is a specific constant, $f^i(x)(i = 1, 2, \ldots, n)$ is a known function with $x$ as its independent variable, and $i$ is the superscript of $f^i$, $f^2 \neq f \cdot f$, $f^3 \neq f \cdot f \cdot f, \ldots, f^n$ and is not equal to the product of $f$.

The mean value of the random function $P(x)$ at a point x is the drift $m(x)$, as:

$$m(x) = E[P(x)] \tag{20}$$

If it is a deterministic function that depicts the spatial trend of the random function $P(x)$, then:

$$Y(x) = P(x) - m(x) \tag{21}$$

where $Y(x)$ is referred to as the residual function of $P(x)$ at $x$.

If $f^0(x_0) = 1$, then the drift at $x_0$ can be expressed as:

$$m(x_0) = a_0 + \sum_{i=1}^{n} a_i f^0(x_0) \tag{22}$$

Thus, there is:

$$\sum_{i=1}^{n} \lambda_i f^0(x_j) = \sum_{j=1}^{k} \lambda_j = f^0(x_0) = 1 \tag{23}$$

Eventually, the mutation function $\gamma(h)$ with only the residual function $Y(x)$ is listed. The pan-Kriging equations for estimating $P(x)$ in existence are as follows:

$$
\begin{aligned}
&\sum_{jj=1}^{k} \lambda_{jj} f^i(x_{jj}) + \mu_0 + \sum_{i=1}^{k} \mu_i f^i(x_j) = \gamma(x_0, x_j), j = 1, 2, \ldots, k \\
&\sum_{j=1}^{k} \lambda_j = 1 \\
&\sum_{j=1}^{k} \lambda_j f^i(x_j) = f^i(x_0), i = 1, 2, \ldots, n
\end{aligned}
\tag{24}
$$

Similarly, the random function $Z(x)$ is also interpolated by the above method.

Approach 2: Natural neighbor interpolation

Spatial interpolation creates continuous surface modeling from discrete sample locations and estimates attribute values. Spatial autocorrelation serves as the foundation for spatial interpolation, which is the closer the distance is, the more similar the objects are in [41].

Spatial autocorrelation is also used in natural neighbor interpolation. Its primary premise is to generate Tyson polygons for all sample locations. When interpolating unknown points, these Tyson polygons will be updated and a to-be-interpolated Tyson polygon will be constructed for unknown points. The sample points in the Tyson polygons that intersect the interpolated Tyson polygons are utilized in the interpolation. The influence weight of the to-be-interpolated Thyson polygon is determined by the intersecting area between the original Thyson polygon and the to-be-interpolated Thiessen polygon, as shown in Figure 7. Formula (31) can be used to represent it:

$$P(x) = \sum_{i=1}^{n} b_i(x) g_i \tag{25}$$

where $P(x)$ is the interpolation result at point $x$, $b_i(x)$ is the weight of interpolation sample points $i(i = 1, 2, \ldots, n)$ with respect to the interpolation point $x$, and $g_i$ is the value at sample point $x$.

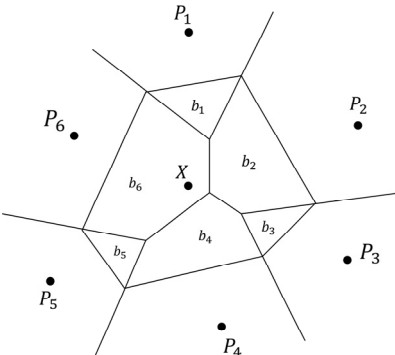

**Figure 7.** Basic principle diagram of natural neighbor interpolation.

The weight can be written using the following formula:

$$b_i(x) = \frac{c_i \cap c(x)}{c(x)}, 0 \le b_i(x) \le 1 \tag{26}$$

where $c_i$ is the area of the Tyson polygon in which the sample point participating in the interpolation is located, $c(x)$ is the area of the Tyson polygon in which the point $x$ to be interpolated is located, and $c_i \cap c(x)$ is the area where the two intersect. Similarly, the random function $Z(x)$ is also interpolated by the above method.

### 3.4. Adaptive Trend Fusion

Following the estimation of the direction of the FC-GLCM for each trend category in the image $\mathcal{L}_f$, the number of pixels contained in each trend category is taken as the confidence of the trend direction, and adaptive trend fusion based on coefficient optimization is performed to obtain the final overall image trend $\varphi^*$ [42]. The process of adaptive trend fusion is shown in Figure 5. The weight of each trend category in the image is optimized using the formula:

$$w_k = \min_{w_k}\left[\|w_k - \widetilde{w}_k\|^2 + \|Z'(\varphi^*)\|^2\right] \tag{27}$$

where $w_k$ is the coefficient of $\varphi_k$ and $\widetilde{w}_k$ is the reference value of $w_k$.

The general trend of the ultimate image is represented by the equation:

$$\varphi^* = \sum_{k=1}^{N_a} \varphi_k \cdot w_k \tag{28}$$

where $N_a$ is the number of dominant groups and $\varphi_k$ is the dominant trend angle of the $k$-th sequence determined using the FC-GLCM.

### 3.5. How to Resolve the 180° Ambiguity Problem

It is challenging to resolve the 180° ambiguity problem in spatial image processing when retrieving sea surface wind direction information. The GLCM method applied in this paper uses a counterclockwise semicircle search method to restrict the angle to 180°, but also has the option to select between a tiny angle and a 180° turnover as other algorithms. The following is a theoretical description of the modified GLCM search strategy, which can effectively solve the 180° ambiguity problem and be applied to other wind direction retrieval methods.

Step 1: Create 360° augmented search domain

Firstly, the 180° search domain of the GLCM is extended to 360°, or from the counterclockwise semicircular search mode to the standard circular search mode. The 360° search range in this study domain extends the angle search range of the GLCM. Because of the mirror relationship between the angles below 180° and the angles reversed by 180°,

the calculation accuracy of the GLCM method will be slightly decreased. The results of the GLCM's wind direction inversion will also be impacted by the unstable calculation accuracy, which makes it difficult to resolve the 180° ambiguity problem.

Step 2: Coupling angle evaluation strategy

While Step 1 overcomes the GLCM method's inability to extract information regarding a wind direction of over 180°, the algorithm's computation accuracy is decreased. This section utilizes a coupling angle determination strategy to address the mentioned problems and further resolve the problem of 180° ambiguity, while simultaneously increasing the calculation accuracy of the algorithm.

Source identification. The source of the sea surface wind direction can be identified based on the fact that the origins between the large and small angles of the wind are mirrored. According to the definition given in this study, the source of sea surface wind is the point at which the local maximum value is within the predominant trend range of the current wind direction data. The likelihood that the wind direction is a small-angle wind direction increases with the proximity of the wind direction source to the origin of the polar co-ordinates. Otherwise, the wind direction is a large-angle wind direction.

Trend analysis. Based on the conclusions of the source identification, the trend of the wind direction is determined as a straight line with three points: the origin of the polar co-ordinates, the source of the wind direction, and the lower boundary of the data, with the source of the wind direction serving as the initial point. The characteristics of small-angle wind are that it begins at the source of the wind and extends to the downward boundary, and the proximity of the source to the origin is another feature. However, the source of the wind direction is closer to the lower border and the trend of the large-angle wind direction extends from the source of the wind direction to the origin of the polar co-ordinates.

The distribution of the P matrix will alter based on the consequence of various trends within the calculating GLCM, which will have a positive impact on the wind direction retrieval result. Figure 8a illustrates that the distribution of the small-angle wind direction ρ in the P matrix is defined by several peaks, which will lead to the general migration tendency. The distribution of φ displays a multi-peak and wide-area trend, an overall left tilt, and evenly spaced pixels. As shown in Figure 8b, the distribution of the large-angle wind direction ρ in the P matrix has a single peak, the characteristic value is rather considerable, and it has a consistent tendency to be non-tilting. The distribution of φ exhibits a single peak, a trend confined to a narrow region, and a right-tilted, steep pixel distribution.

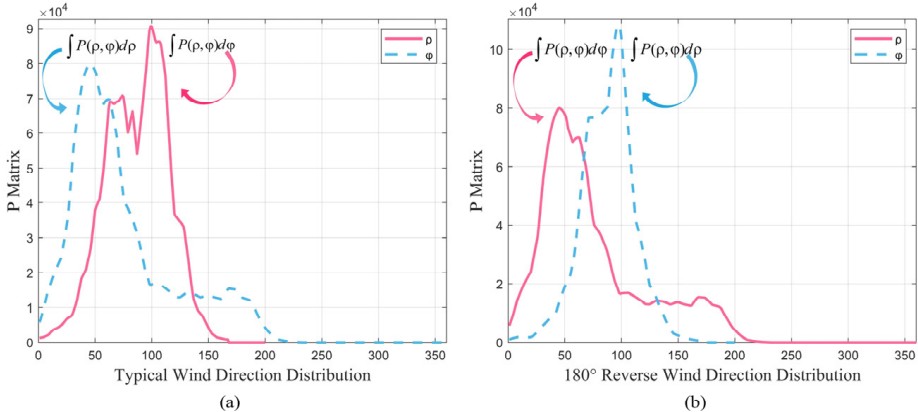

**Figure 8.** P matrix distribution at different angles. (**a**) Distribution of the small-angle wind direction; (**b**) distribution of the large-angle wind direction.

In summary, firstly, the feature distribution direction and position of the wind direction are judged in the process of extracting the dominant features in the preprocessing part for the input radar image. Secondly, the characteristic change trend of the wind direction is

evaluated. Finally, utilizing the distinctive distribution direction and location of the wind direction that will cause the distribution of the P matrix to change, the final estimated sea surface wind direction can be accurately determined in GLCM calculations. Figure 5 is taken as an example to illustrate this principle, where the current reference wind direction is 48°. In terms of characteristic distribution direction and wind direction, the characteristic distribution in Figure 5 is characterized by a multi-segment distribution, with a general left-leaning dominant trend. This distribution mode will cause the P matrix to have multi-peak characteristics, resulting in an overall migration tendency. The change trend in Figure 5 is relatively gentle, with an obvious transition trend. As a result, during the feature extraction process, the result frequently has the feature of edge augmentation and there will be a certain number of edge transition zones. Due to the smooth pixel distribution of the wind direction, the P matrix distribution presents a trend of multi-peak and large areas, as shown in Figure 8a. Due to different characteristic distribution directions and positions, their peak positions will also be different, as shown in the peak position of the curve in Figure 7. In addition, since the characteristic change trend of the wind direction in Figure 5 is relatively gentle, the peak value of its P matrix distribution will also produce more secondary peaks, as shown in the pink curve in Figure 8a.

## 4. Experiments Results

In this section, we will analyze 1436 sets of data that were collected between 22 October and 30 October 2010. This dataset includes a wind speed range between slightly higher than 2 m/s and 20 m/s, a wave direction range between 65° and 257°, and a wave height range between 1 m and 3 m. As evaluation criteria, Kullback–Leibler divergence (KLD), root-mean-square error (RMSE), and Pearson's correlation coefficient (PCC) are chosen to compare the FC-GLCM, natural neighbor, Kriging, traditional GLCM (T-GLCM), the adaptive reduced method (ARM), and the energy spectrum method (ESM).

KLD divergence measures the divergence between the distribution $S$ and $\hat{S}$:

$$KL(S, \hat{S}) = \sum_{i=1}^{N \times N} \hat{S}_i \log \frac{\hat{S}_i}{S_i} \tag{29}$$

where $S$ represents the measurement speed data by vane, $\hat{S}$ represents the wind direction data estimated by the algorithm, and $N \times N$ represents the matrix of the input radar image.

RMSE reflects the extent to which $\hat{S}$ deviates from $S$:

$$S_{RMSE} = \sqrt{\frac{\sum_{i=1}^{N \times N} (S_i - \hat{S}_i)^2}{N}} \tag{30}$$

PCC measures the linear correlation between $S$ and $\hat{S}$:

$$PCCs = \frac{\sigma(S, \hat{S})}{\sigma(S)\sigma(\hat{S})} \tag{31}$$

where $\sigma(S, \hat{S})$ is the covariance of $S$ and $\hat{S}$.

### 4.1. Sensitivity Analysis

The preparation of the polar co-ordinates' sea surface static images is carried out in the third section of the paper, which includes the Gaussian difference filtering procedure with the two parameters $k_1$ and σ. The number of iterations $\xi$ plays a crucial role in the calculation accuracy and speed when using the FC-GLCM algorithm. We analyze the sensitivity of these three parameters. Tables 2 and 3 demonstrate that when $k_1$ and σ increase, KLD and RMSE show a pattern of decreasing first and then increasing, whereas PCC shows a trend of increasing first and then decreasing. KLD and RMSE are the smallest and PCC is the biggest when $k_1 = 1.60$ and σ $= 0.80$, respectively. This shows that

the distribution of the wind direction retrieved by the FC-GLCM method is almost the same as that of the reference wind direction, with the lowest standard deviation and best correlation. Figures 9 and 10 demonstrate that the retrieved wind direction will be larger than the reference wind direction when $k_1$ and σ are too large or too small, but when $k_1$ is excessively large, the estimated wind direction will be about twice as large as the reference wind direction. When σ is excessively large, the estimate is approximately double the reference value. Table 4 illustrates the variations in KLD, RMSE, and PCC with increasing iterations. As shown in Figure 11, KLD and RMSE are minimum and PCC is maximum when the number of iterations $\xi = 3$. In conclusion, the algorithm performs best when $k_1 = 1.60$, σ = 0.80, and $\xi = 3$.

**Table 2.** Sensitivity analysis of parameter $k_1$.

| $k_1$ | 0.16 | 0.64 | 1.12 | 1.60 | 1.76 | 1.88 | 16.00 |
|---|---|---|---|---|---|---|---|
| KLD | 0.9742 | 0.8788 | 0.7600 | 0.6954 | 0.7380 | 2.3865 | 4.5658 |
| RMSE (°) | 11.0756 | 8.4816 | 5.0022 | 4.3236 | 4.3669 | 47.3704 | 92.8893 |
| PCC | 0.0289 | 0.0288 | 0.0291 | 0.0301 | 0.0293 | 0.0291 | 0.0292 |

**Table 3.** Sensitivity analysis of parameter σ.

| σ | 0.08 | 0.32 | 0.56 | 0.80 | 0.88 | 4.40 | 8.00 |
|---|---|---|---|---|---|---|---|
| KLD | 0.8758 | 0.8294 | 0.7839 | 0.7277 | 0.7652 | 1.5085 | 2.4246 |
| RMSE (°) | 7.6117 | 6.3330 | 5.2913 | 5.2017 | 5.3398 | 25.3538 | 48.0796 |
| PCC | 0.0293 | 0.0293 | 0.0291 | 0.0295 | 0.0291 | 0.0295 | 0.0292 |

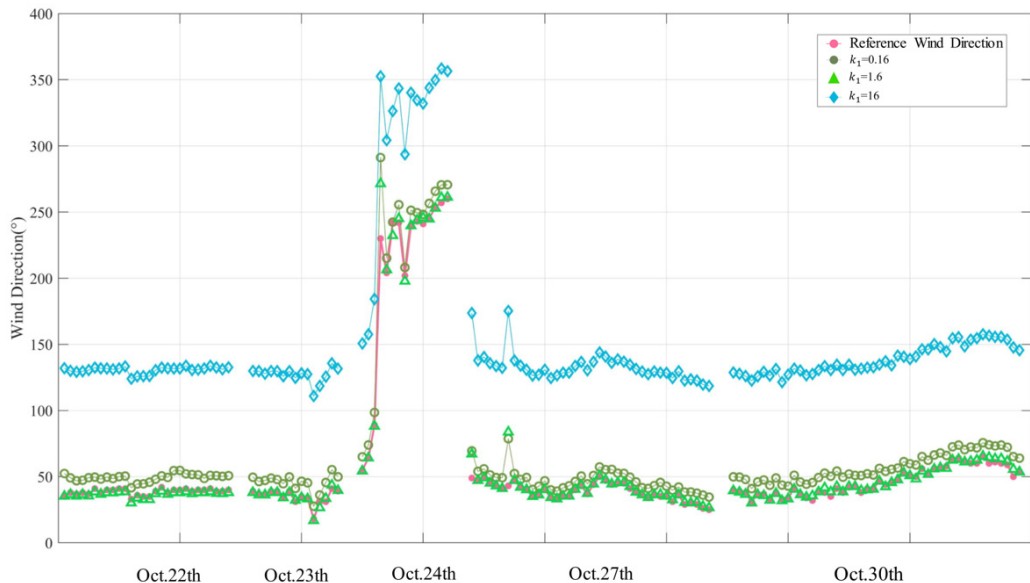

**Figure 9.** Wind direction sequence obtained by FC-GLCM algorithm when parameter $k_1$ is different. (The pink curve with the open points represents the wind direction measured by the vane anemometer.)

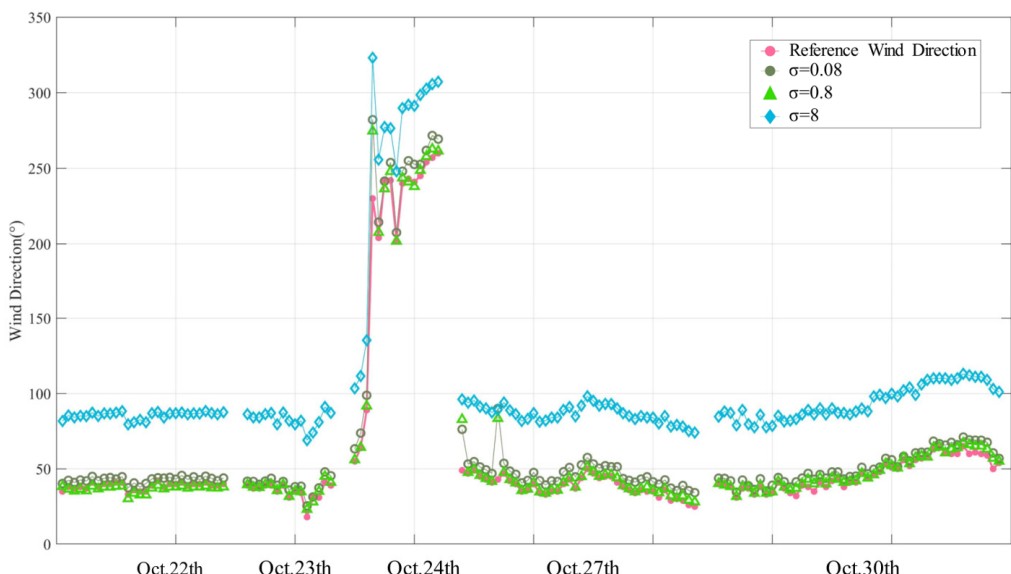

**Figure 10.** Wind direction sequence obtained by FC-GLCM algorithm when parameter $\sigma$ is different. (The pink curve with the open points represents the wind direction measured by the vane anemometer.)

**Table 4.** Sensitivity analysis of iteration number $\xi$.

| $\xi$ | 1 | 2 | 3 | 4 | 5 | 6 | 7 |
|---|---|---|---|---|---|---|---|
| KLD | 0.3239 | 0.3153 | 0.2856 | 0.3240 | 0.3227 | 0.3216 | 0.3253 |
| RMSE (°) | 4.7254 | 4.7653 | 4.6455 | 4.7242 | 4.7284 | 4.7212 | 4.7193 |
| PCC | 0.0295 | 0.0294 | 0.0305 | 0.0292 | 0.0292 | 0.0292 | 0.0294 |

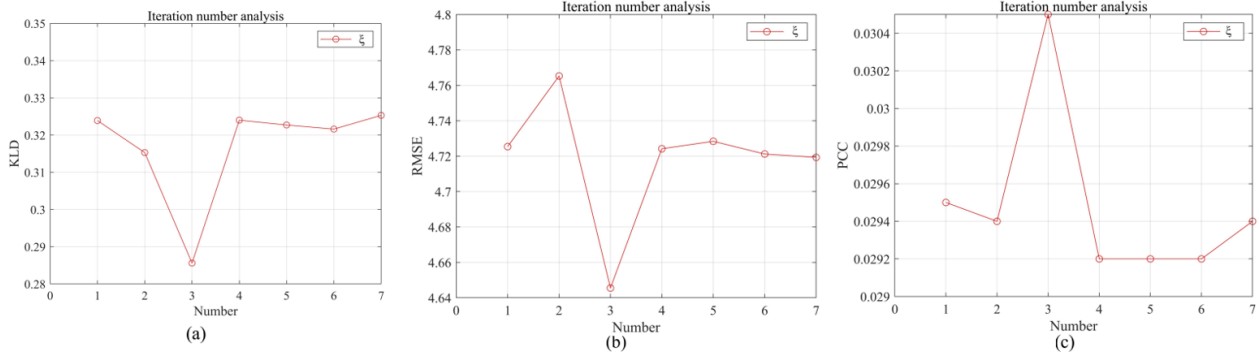

**Figure 11.** Wind direction sequence obtained by FC-GLCM algorithm when iteration number $\xi$ is different. (**a**) KLD index; (**b**) RMSE index; (**c**) PCC index.

### 4.2. Accuracy Validation

The ARM, ESM, and T-GLCM algorithms are compared with the FC-GLCM algorithm, the Kriging interpolation method, and the natural neighbor interpolation method. The results, evaluation criteria, and computing time are shown in Table 5. The table shows that for the retrieved wind direction data compared to the data measured by the in situ vane anemometer, the KLD of the FC-GLCM algorithm proposed in this paper has 0.6404, and the RMSE is 4.9867°. The correlation coefficient is as high as 0.9268. Table 5 demonstrates that the FC-GLCM reduces the computation time by 63.8%, 50.6%, and 42.9%, respectively, compared to ARM, ESM, and T-GLCM, which has the fastest calculation speed. In addition, in terms of the KLD indicator, the FC-GLCM reduced by 0.05%, 32.7%, and 16.8%, respectively, when compared to T-GLCM, ARM, and ESM. In terms of the RMSE indicators, the FC-GLCM decreased by 18.1%, 91.6%, and 67.7%, respectively, compared to T-GLCM,

ARM, and ESM. When compared to the FC-GLCM method, the RMSE of Kriging and natural neighbor interpolation increases by 0.4616° and 0.3902°, respectively. Nevertheless, the calculation time is decreased by 21% and 26%, respectively. The method proposed in this study may efficiently increase the accuracy and calculation speed of the T-GLCM algorithm, and the proposed interpolation method can also greatly enhance the calculation speed with minimal accuracy cost.

**Table 5.** Comparison of the proposed FC-GLCM method with classic wind measurement technique.

| Method | KLD | RMSE (°) | PCC | Computing Time(s) |
|---|---|---|---|---|
| ARM | 0.9516 | 59.3512 | 0.3112 | 25.2679 |
| ESM | 0.7699 | 15.4349 | 0.9122 | 18.4831 |
| FC-GLCM | 0.6404 | 4.9867 | 0.9268 | 9.1391 |
| Kriging | 0.6137 | 5.4483 | 0.9267 | 7.2608 |
| Natural Neighbor | 0.6182 | 5.3769 | 0.9267 | 6.7746 |
| T-GLCM | 0.6713 | 6.0533 | 0.9265 | 15.9731 |

Figure 12 shows the line diagram of each algorithm divided according to time sequences under the PCC indicator. The data under weak rainfall are shown in the gray background region of the figure. Figure 12a illustrates how poorly the ARM algorithm performs in the PCC indicator and its retrieved wind direction information differs greatly from the reference wind direction data in all cases, especially in rainy weather when its error can reach more than 80%. The gray background area in Figure 12 represents the weak rainfall data. The red dotted boxes represent the data with noticeable jitter during weak rainfall. The red dotted box on the right shows the weak rainfall data from 13:12 to 19:30 on 30 October. During this period, the RMSE of the FC-GLCM was 5.0167°, which was 94.5% and 82.7% lower than that of ARM and ESM, respectively. The RMSE of the Kriging interpolation and natural neighbor interpolation proposed in this paper is 5.4248° and 5.2356°, respectively. The left red dotted box shows the rainfall data from 1:22 to 6:37 on 24 October. At this time, the sea surface wind direction is a large-angle wind direction. The RMSE of the FC-GLCM is only 0.26°, which is 88.2% lower than that of ARM and ESM. The RMSE of the two interpolation methods proposed in this paper is 1.16°. Therefore, the FC-GLCM algorithm can not only effectively resist the impact of rainfall on the retrieval of sea surface wind direction information using the Gaussian filtering and feature enhancement methods in the preprocessing part but can also accurately identify the wind direction at all angles using the different spatial distributions of the P matrix.

Figure 13 shows the two-dimensional distribution of the KLD indicators in the sample data. The closer the KLD is to zero, the more similar the distribution of the reference and estimated sea surface wind directions. The more accurate the model, the more accurate the wind direction inversion. Figure 13a demonstrates that the FC-GLCM algorithm is distributed at 0~2, the ARM is distributed at −4~10, the ESM is distributed at −2~4, T-GLCM is distributed at −3~4, and both interpolation algorithms are distributed at −2~3. Therefore, the FC-GLCM algorithm proposed in this paper has the most concentrated distribution and the algorithm has higher accuracy and better convergence in retrieving sea surface wind direction information.

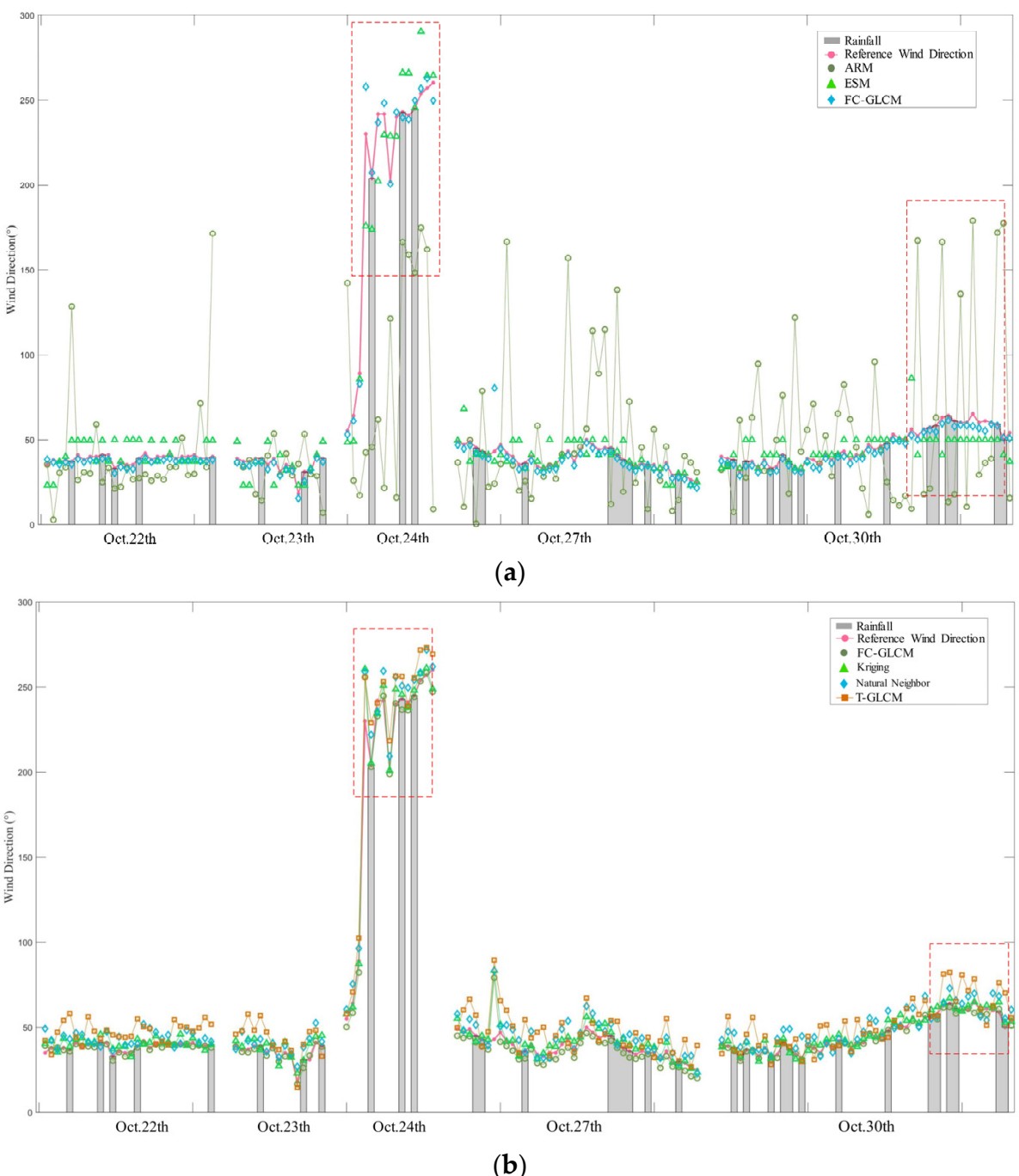

**Figure 12.** Comparison of the proposed FC-GLCM method results with different wind measurement techniques. (The pink curve with the open points represents the wind direction measured by the vane anemometer, the gray background area is the data during weak rainfall, and the red dotted boxes represent the data with large error during rainfall.) (**a**) Comparison of the proposed FC-GLCM method results with the Kriging, natural neighbor, and T-GLCM approaches. (**b**) Comparison of the proposed FC-GLCM method results with the ARM, ESM.

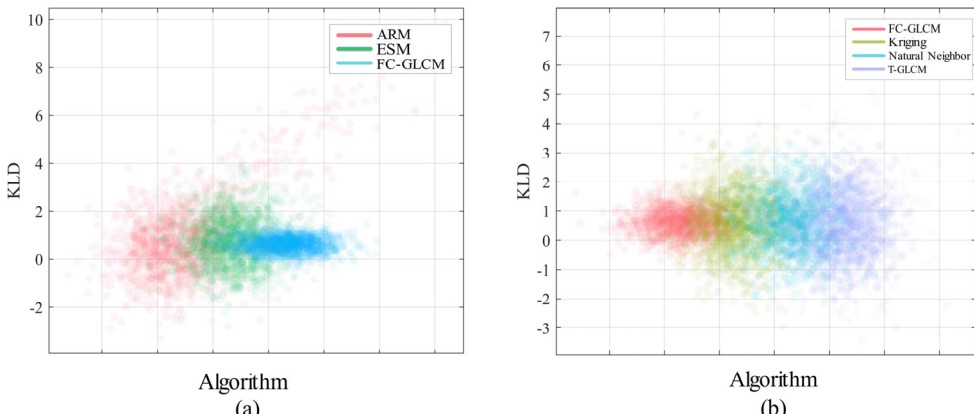

**Figure 13.** Comparison of KLD index between the proposed FC-GLCM and classic algorithm. (**a**) Comparison of the proposed FC-GLCM method with the ARM and ESM; (**b**) comparison of the proposed FC-GLCM method with the Kriging, natural neighbor, and T-GLCM approaches.

Figure 14 shows the three-dimensional distribution of the sample data RMSE indicators. Table 5 shows that the RMSE of the FC-GLCM algorithm is only 4.9867°, while that of ARM is 59.3512°. The distribution of the FC-GLCM algorithm is more centralized, as shown in Figure 14a, while the distribution of the ARM is the most decentralized. It can be seen from Figure 14b that the convergence of the two interpolation algorithms is relatively weak compared with T-GLCM, but the RMSE of both algorithms is lower than T-GLCM according to Table 5. Therefore, it shows that the two interpolation methods will reduce the robustness and improve the performance of the improved algorithm.

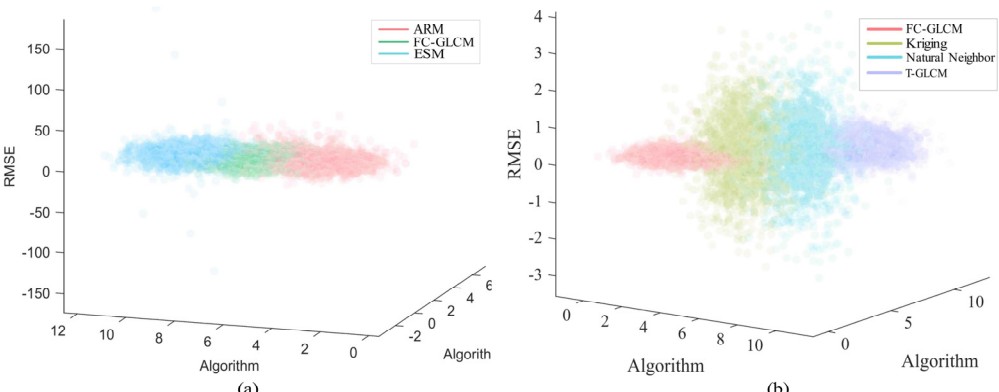

**Figure 14.** Comparison of RMSE index between the proposed FC-GLCM and classic algorithm. (**a**) Comparison of the proposed FC-GLCM method with the ARM and ESM; (**b**) comparison of the proposed FC-GLCM method with the Kriging, natural neighbor, and T-GLCM approaches.

### 4.3. Robustness Test

This section redistributes the reference data used in this study according to various sea conditions and investigates how these factors affect the algorithm's accuracy. Weak wind, strong wind, wave height, wave direction, and rainfall are among the sea condition parameters tested in this section.

Table 6 shows how each algorithm's accuracy decreases in response to strong wind conditions. The FC-GLCM algorithm in this paper not only has significantly greater accuracy than the other algorithms, but also has a lower RMSE compared to the other algorithms. Considering the influence of wind speed on the retrieval of sea surface wind direction information, this paper divides the sample data into two categories: small wind speed (2–13.5 m/s) and large wind speed (13.5–20 m/s). There are 746 groups of data for small wind speed and 690 groups of data for large wind speed. Figure 15 shows the wind

direction error between the estimated wind direction and the reference wind direction at different wind speeds. The shaded area is the wind speed data measured by the vane. Figure 15a shows the error between the estimated wind direction and the reference wind direction when the wind speed is 2–13.5 m/s. The red dotted boxes represent the area with large fluctuations in the error range. The large error range of the left dotted box is caused by the large fluctuation in the wind speed between 7–11 m/s and the large angle wind direction at this time. At the time, the RMSE of ARM can reach 71.2691°, while the RMSE of the FC-GLCM algorithm is only 9.7285°. The dotted box on the right is the error data when the wind speed is approximately 12 m/s. The estimated wind direction is basically consistent with the reference wind direction and the RMSE of the FC-GLCM is only 2.7487°, which is 42.7512° and 21.4726° less than the RMSE of ARM and ESM, respectively. The red dotted box in Figure 15b shows the error data when the wind speed is 13–14 m/s. At this time, the RMSE of the FC-GLCM is only 2.3565°, which is 94.9% and 64.9% lower than ARM and ESM, respectively. ARM and ESM have a larger error range when the wind speed is approximately 13 m/s. However, the fast convergence and adaptive fusion characteristics of the FC-GLCM ensure the stability, robustness, and accuracy of the retrieved results.

**Table 6.** Comparison between FC-GLCM algorithm and classic method under different sea conditions.

| Sea Condition | Indicator | ARM | ESM | T-GLCM | FC-GLCM |
|---|---|---|---|---|---|
| Weak Wind | KLD | 0.9361 | 0.7735 | 0.6054 | 0.4518 |
| | RMSE (°) | 74.7396 | 19.7976 | 6.9043 | 4.4738 |
| | PCC | 0.3545 | 0.9140 | 0.9252 | 0.9277 |
| Strong Wind | KLD | 0.9670 | 0.7655 | 0.6114 | 0.5124 |
| | RMSE (°) | 35.8954 | 8.4672 | 6.0449 | 4.4738 |
| | PCC | 0.2081 | 0.3978 | 0.8289 | 0.8395 |
| Wave Height | KLD | 0.8931 | 0.7906 | 0.5987 | 0.4523 |
| | RMSE (°) | 56.4481 | 10.7277 | 6.3990 | 5.2223 |
| | PCC | 0.4723 | 0.9158 | 0.9169 | 0.9272 |
| Wave Direction | KLD | 0.8931 | 0.7906 | 0.5987 | 0.4523 |
| | RMSE (°) | 56.4481 | 10.7321 | 6.3979 | 5.2222 |
| | PCC | 0.4733 | 0.9158 | 0.9169 | 0.9272 |
| Rainfall | KLD | 2.3897 | 0.6503 | 0.6445 | 0.5579 |
| | RMSE (°) | 91.2342 | 29.0755 | 6.5193 | 5.0167 |
| | PCC | -0.0322 | 0.8569 | 0.9061 | 0.9263 |

Since the wave height and wave direction data are averaged for 20 min, in order to analyze the time correspondence, it is also necessary to average the retrieval results for 20 min. There are 170 sets of data after averaging. Table 6 indicates that the wave height and wave direction will indeed have a negative impact on the algorithm's accuracy and robustness when they are significantly changed. Figure 16 shows the wind direction error between the retrieved direction data and reference wind direction data at different wave heights. Figure 17 shows the wind direction error between the retrieved direction data and the reference wind direction data at different wave directions. It can be seen from Table 6 that the RMSE of the overall sample of the FC-GLCM is 5.2223° when the wave height is different, which is reduced by 51.2258°, 5.5024°, and 1.1767°, respectively, compared with ARM, ESM, and T-GLCM. The red dotted box in Figure 16 shows the data with large error fluctuations and the wave height is about 1.5 m. In this area, the sample RMSE of the FC-GLCM, ESM, and ARM is 3.60°, 3.82°, and 26.53°, respectively. The large error range of the dotted line box in Figure 17 is caused by the large wave direction, which reaches around 180°. Compared with ARM and ESM, the FC-GLCM still has higher precision, and the sample RMSE of the FC-GLCM is only 3.72°. To sum up, both ARM and ESM have larger errors due to the influence of wave height and direction, while the FC-GLCM greatly suppresses the influence of such sea conditions on inversion results due to the feature extraction and dominant wind direction data acquisition steps in its preprocessing.

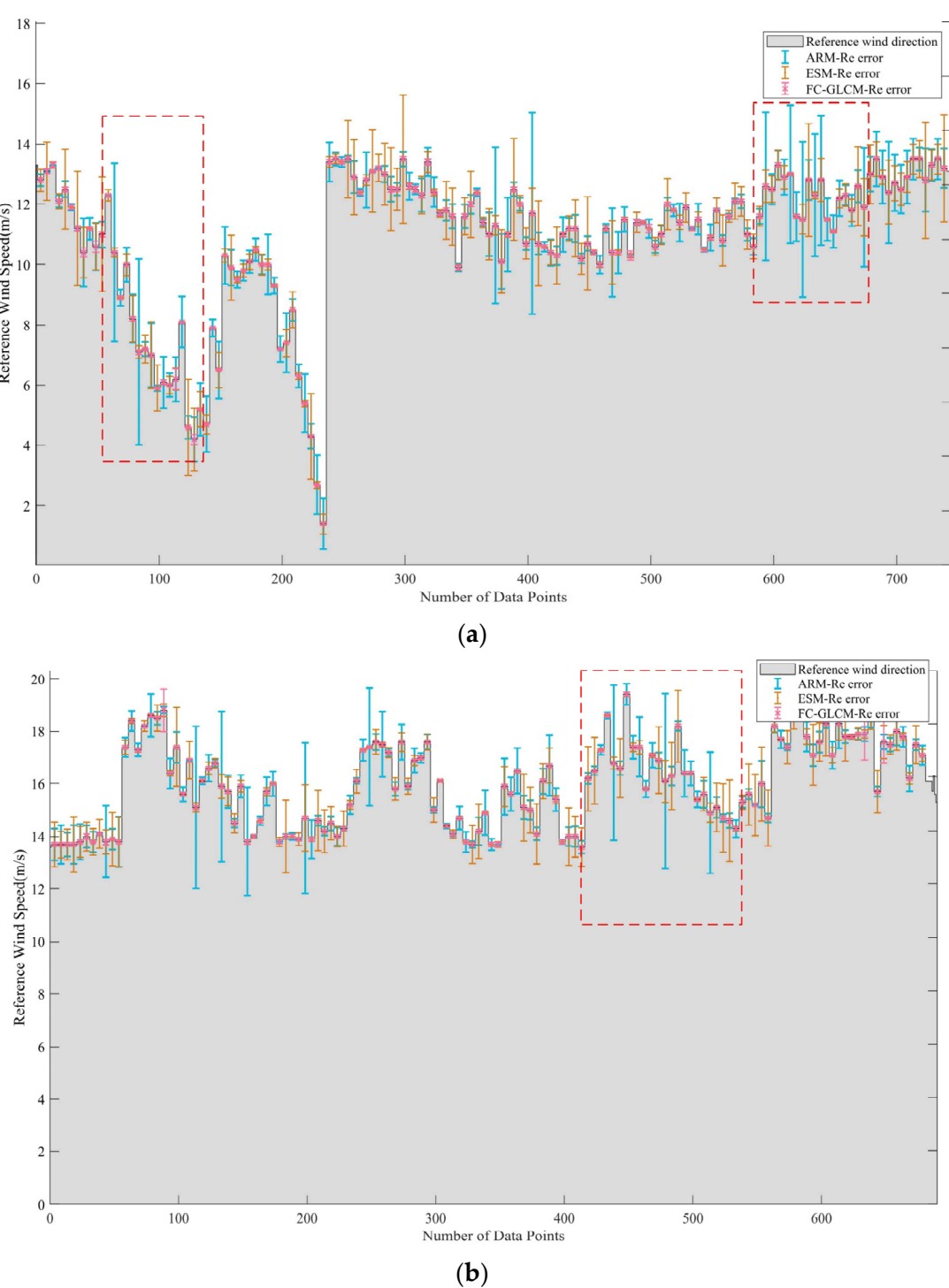

**Figure 15.** Wind direction error between estimated direction and reference wind direction at different wind speeds. (The shaded area represents measurement speed data by vane and the red dotted box represents the area with large fluctuations in the error range.) (**a**) Wind direction error between retrieved direction and reference wind direction when the wind speed is 0–13 m/s. (**b**) Wind direction error between retrieved direction and reference wind direction when the wind speed is 13–20 m/s.

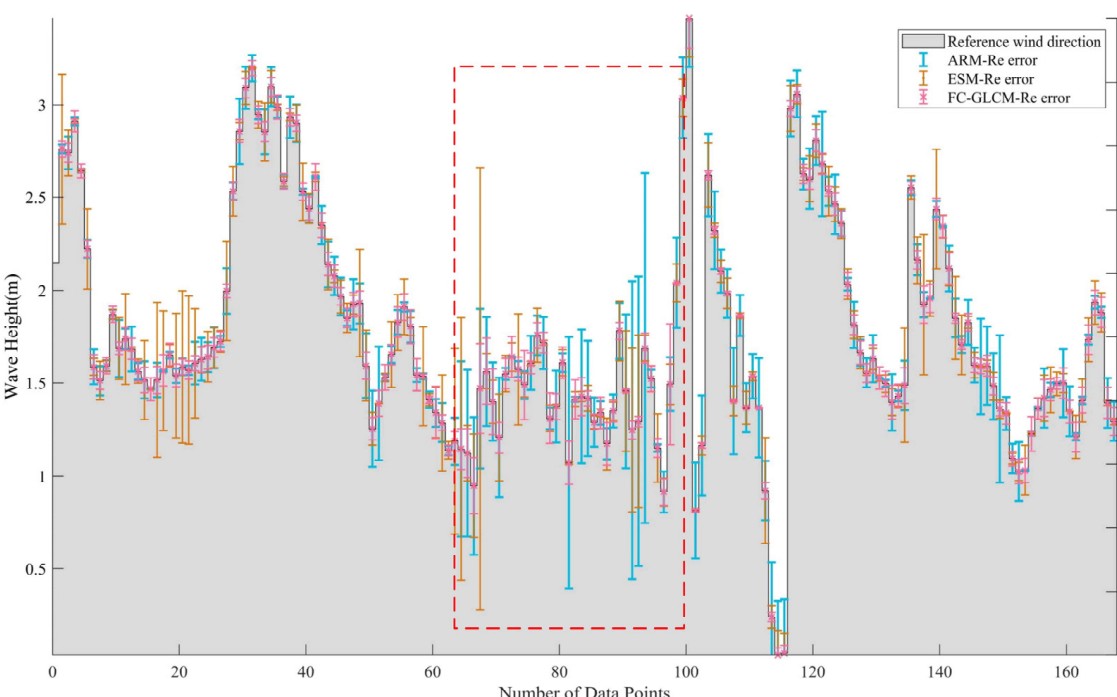

**Figure 16.** Wind direction error between retrieved direction and reference wind direction at different wave heights. (The shaded area represents wave height data measured by vane and the red dotted box represents the area with large fluctuations in the error range.)

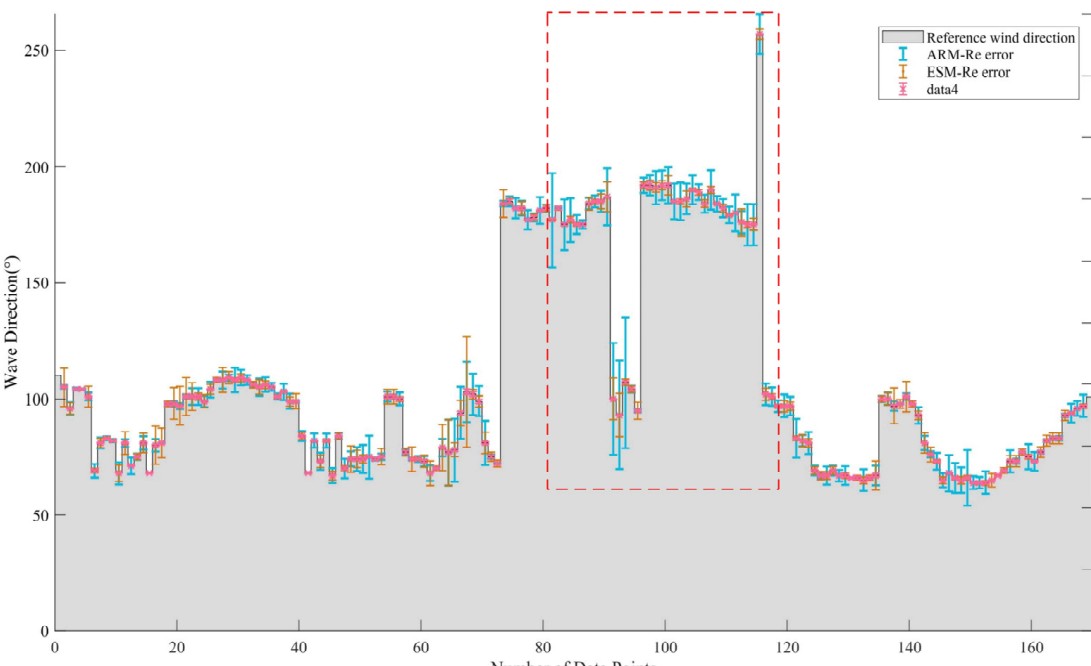

**Figure 17.** Wind direction error between retrieved direction data and reference wind direction data at different wave directions. (The shaded area represents wave direction data measured by vane and the red dotted box represents the area with large fluctuations in the error range.)

In this section, the algorithm is verified by the above series of experiments. Firstly, the sensitivity of variable parameters in the FC-GLCM is analyzed. The experimental results show that the algorithm accuracy will vary with the filter variance $\sigma$ and variance coefficient $k_1$, which are convex set distributions near the optimal parameters, and the highest precision

is in the optimal parameter combination $\sigma = 0.16$ and $k_1 = 0.80$. Increasing the number of iterations improves the accuracy to some extent, while increasing the number of calculations. The number of iterations chosen is $\xi = 3$ based on accuracy and speed. Secondly, for the 1436 groups of measured sea surface wind direction data, this section compares the FC-GLCM algorithm with other algorithms. The FC-GLCM algorithm has the highest accuracy. Its correlation coefficient with the reference sea surface wind direction data is up to 0.9268 and the RMSE is 4.9867°. Finally, this section also compares the FC-GLCM with other classical algorithms for data under various sea conditions. At different wind speeds, the RMSE of FC-GLCM is 4.4738°. At different wave heights and wave directions, the RMSE of the FC-GLCM is 5.2223°. Under rainfall conditions, the RMSE of the FC-GLCM is 5.1067°. Therefore, the FC-GLCM algorithm proposed in this paper has the best accuracy and robustness under different sea conditions.

## 5. Conclusions

In this paper, a new method is proposed for retrieving the sea surface wind direction information from X-band radar image sequences. First, our method proposes an FC- GLCM algorithm based on angle iteration, which improves computation speed and optimizes the GLCM accuracy range. This paper also proposes the Kriging and natural neighbor interpolation methods, which ensure that the FC-GLCM satisfies the requirements of speed and accuracy to better meet the needs of practical engineering. Furthermore, expanding the traditional GLCM algorithm in polar co-ordinates and enabling GLCM to be used with the whole static feature images of the sea surface not only improves the accuracy of the retrieved data, but also simplifies the complexity of the algorithm flow. Finally, the adaptive trend fusion and image preprocessing process of the algorithm solve the bottleneck problem (180° ambiguity problem) associated with the existing inversion method, which can ensure the real-time engineering application of the algorithm and can also be transplanted to other algorithms to solve the problem. In comparison to ARM, ESM, T-GLCM, Kriging, and natural neighbor in the 1436 sets of measured wind direction datasets, the proposed method obtains a KLD of 0.6404, an RMSE of 4.9867°, and a correlation coefficient of 0.9268. The paper compares the FC-GLCM with ESM and ARM for the data under different sea conditions after categorizing the measured sea surface wind direction data for sea conditions. Under different sea circumstances, the FC-GLCM produced the best accuracy, with an average accuracy reduction rate of only 8%. Especially when rainfall has a great impact on image quality, the accuracy reduction rate of the FC-GLCM is less than 13%. This paper demonstrates the engineering feasibility of the new method from both theoretical analysis and experimental data. Compared with the traditional GLCM, the accuracy and robustness of the new method are improved and the calculation speed is also guaranteed. However, the sea conditions investigated in this paper are still insufficient for investigating application areas and there is a scarcity of algorithmic application research under harsh conditions. Moreover, the algorithm proposed in this article is not applicable in situations where wind speed is extremely low. In further study, we will optimize the algorithm's performance in extreme situations and broaden the application environment for the GLCM in sea surface wind direction inversion.

**Author Contributions:** The paper is provided by seven authors; the authors' contributions are as follows: conceptualization, H.W.; data curation, Z.L.; funding acquisition, Z.Z. and H.G.; methodology, H.W.; project administration, Z.Z.; software, H.Q.; supervision, Y.W.; writing—original draft, S.L.; writing—review and editing, H.W. All authors have read and agreed to the published version of the manuscript.

**Funding:** This research was funded by National Natural Science Foundation of China (41906154, 52101358), Jiangsu Provincial Key Research and Development Program Social Development Project (BE2022783), and the Zhenjiang Key Research and Development Plan (Social Development) Project, No. SH2022013.

**Data Availability Statement:** Not applicable.

**Acknowledgments:** The authors would like to thank the two anonymous reviewers for their valuable comments and suggestions.

**Conflicts of Interest:** The authors declare no conflict of interest.

## Appendix A

**Theorem A1.** *The dominant trend angle $\varphi^*$ for a given image $\mathcal{L}_f$ is calculated using the FC-GLCM algorithm. The dominance of its trend may be evaluated using parameters $\mathcal{K}$ measurement, and it is unique.*

**Proof of Theorem A1.** Assuming that parameter $\mathcal{K}$ is a parameter measuring the trend size of pixels in image $\mathcal{L}_f$, the size of parameter $\mathcal{K}$ by the ratio of the number of trend pixels $N_e$ to the number of other pixels $N_0$ (the set of all pixels in the image $\mathcal{L}_f$ except trend pixels) and its relationship follows the below formula:

$$\mathcal{K} \propto \frac{N_e}{N_0} \tag{A1}$$

Other pixel sets are only uniquely defined by the trend pixel set when given the whole set of current pixels.

The pixel point $\mathcal{L}_f(r_i, \theta_j)$ in the image $\mathcal{L}_f$ is uniquely defined by the coordinate pair $(r_i, \theta_j)$, where $i \leq M, j \leq N$, $M$ is the number of rows in the image $\mathcal{L}_f$, and $N$ is the number of columns in the image $\mathcal{L}_f$, where image is $\mathcal{L}_f \in \mathbb{R}^{M \times N \times 3}$. The following definition applies to the trend pixel set $\mathcal{G}$:

$$\mathcal{G}(\varphi, \eta) = \{(r, \theta) | \Phi(\psi((r, \theta) : a) | \mathcal{F}(a)) > \eta\} \tag{A2}$$

where $\psi((r, \theta))$ is the coordinate pair $(r, \theta)$ for the circle's center, and the bottom semicircle is uniquely defined for the search radius $\rho$. In the search region $a$, $\psi((r, \theta) : a)$ denotes the pixel group determined by the distribution angle of the most advantageous related pixels. $\mathcal{F}(a)$ is the area-limiting function, which restricts the coordinate pairs $(r, \theta)$ in the search area $a$ to the image's boundaries to prevent out of bounds. $\Phi(\psi((r, \theta) : a) | \mathcal{F}(a))$ is a pixel ratio in the whole $a$ determined by calculation $\psi((r, \theta))$. $\eta$ denotes the ratio threshold, which is uniquely determined by the image $\mathcal{L}_f$ and can be written using the formula:

$$\eta = \mu \pm 2\sigma \tag{A3}$$

where $\sigma$ and $\mu$ denote function $\Phi$, the standard deviation and mean value of all pixels in the image $\mathcal{L}_f$ after the procedure. $\square$

The feature of the trend $\varphi$ in image $\mathcal{L}_f$ has been more distinctive when the set of trend pixels $\mathcal{G}(\varphi, \eta)$ in the image $\mathcal{L}_f$ contains more pixels, and the value of the parameter $\mathcal{K}$ has been greater. The parameter $\mathcal{K}$ can be used to determine the dominant degree of the trend when the image $\mathcal{L}_f$ has a unique dominant angle $\varphi^*$. We have demonstrated that the radar image processed with FC-GLCM have a distinct optimal solution $\varphi^*$ in Theorem A1. Theorem A2 will give a detailed insight into the convergence behavior of the FC-GLCM.

**Theorem A2.** *The FC-GLCM algorithm converges to any image $\mathcal{L}_f$, with the convergence rate parameter $\xi$ controlling the rate of convergence.*

**Proof of Theorem A2.** For the $k$-th gray-level co-occurrence matrix operation angle sequence $\varphi_k$ and its update iteration result sequence $\varphi_{k+1}$, the advantage trend angle $\varphi^*$ of the sequence $\varphi_k$ is $\varphi_{k+1} = U(\varphi_k^*, \varepsilon_{k+1})$ according to the operation law of the update

iteration mode. The relationship between the gray-level co-occurrence matrix operation result and the corresponding angle sequence $Z'(\varphi)$ satisfies the formula:

$$Z'\left(\varphi_{k+1}^*\right) \leq K\varepsilon_{k+1} Z'\left(\varphi_k^*\right) \tag{A4}$$

In addition, when $K\varepsilon_{k+1} \leq 1$, with $Z'\left(\varphi_{k+1}^*\right) > 0$ is true for $\forall k > 0, k \in \mathbb{Z}$, then $\exists \lambda < 1$ makes:

$$\left|Z'\left(\varphi_{k+1}^*\right) - Z'\left(\varphi_k^*\right)\right| < \lambda Z'\left(\varphi_k^*\right) \leq \xi Z'\left(\varphi_0^*\right) \tag{A5}$$

where $\varphi_0^*$ is the initial angle sequence, and $\exists \mathcal{M} \in \mathbb{Z}$ means that $Z'\left(\varphi_0^*\right) \leq \mathcal{M}$ is established.

For $\forall k > 0, k \in \mathbb{Z}$, $K\varepsilon_{k+1} \leq 1$ is formed, and $\lambda < 1$, $\xi = \lambda \prod_{\rho=1}^{k} K\varepsilon_\rho < 1$ is established.

Further, there is

$$\left|Z'\left(\varphi_{k+1}^*\right) - Z'\left(\varphi_k^*\right)\right| < \mathcal{M} \tag{A6}$$

Then, the sequence $\{Z'(\varphi)\}$ must be convergent, and the convergence rate is controlled by the convergence parameter $\xi$. $\square$

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
