# Peer review of "Development of a Fast Convergence Gray-Level Co-Occurrence Matrix for Sea Surface Wind Direction Extraction from Marine Radar Images"

_remotesensing, doi:10.3390/rs15082078_

Round 1

Reviewer 1 Report

Comments and Suggestions for Authors

In this study, the new sea surface wind direction from the X-band marine radar image is proposed using a fast convergent gray-level co-occurrence matrix (FC-GLCM) algorithm. The MRMDL method provides a new theoretical and methodological reference for retrieving sea surface wind direction from ocean radar images. However, the publication can only be recommended after major revision and solving the following problems.

1.       Introduction. The introduction is too long for the reader to capture key information. I strongly suggest that it be simplified.

2.       In Section 3, readers may notice that there are many formulas involved in the paper, and opinions can be presented as supplementary materials.

3.       Figure 8 is very vague. Other figures also have this problem. Please modify it.

4.       There is a typing error in line 428, please correct it.

5.       I suggest providing data acquisition links in the data description section to facilitate readers' access to data.

6.       I can notice from the paper that the focus of the author's research is the retrieval of sea surface wind direction. However, the accuracy of wind direction retrieval is usually affected by many factors, such as wind speed, rainfall, sea waves, etc. Can the author clarify in the paper how to solve the retrieval of sea surface wind direction in the case of wind speed, rainfall and sea wave driving changes.

Author Response

Thank you very much for the expert's opinion, all the opinions are very professional, have a lot of help for improving the level of the paper.

Please refer to the attached PDF for the authors' response to the reviewer comments. 

Reviewer 2 Report

In this manuscript, a novel method is proposed to retrieve wind direction from X-band marine radar images using a fast convergent GLCM features algorithm. The methodology is written in detail and the experimental results are presented comprehensively. Although the method is new to the wind retrieval research in marine radar images, the current version is not suitable for publication unless the following issues are addressed and corrected.

  1. Line 22 and 23, the values of correlation coefficients and RMSE do not make sense. It is impossible to get such nearly 100% accuracy unless there are some mistakes in methodology, reference data, or statistical calculations. As it is impossible for reference wind data collected from the anemometer to have a measurement resolution of 0.001 degree, it does not make sense to obtain a RMSE of 0.0018 degree. 

  2. Line 26, the decreases in estimation error compared to previous methods do not make sense as well. 

  3. line 27, 180 degree fuzzy problem should be changed to 180 degree ambiguity problem.

  4. line 30, add “estimation” after “sea surface wind”.

  5. line 35, change the order between “air flow distortion” and “inaccurate estimation”.

  6. line 38, “Although these methods” should be “Although these sensors”.

  7. line 39, the authors mention SAR has high resolution but cannot conduct small area measurements, which sounds contradictory to each other.

  8. line 43, remove “detection” after “rain”.

  9. line 44, “more” should be “most”.

  10. line 56, it does not look formal in academic writing to start the sentence with “[reference number] proposed…”, the authors of the reference should be included. Please check and correct this issue in all the text in the manuscript.

  11. line 66, “the authors” looks vague and should be replaced with the name (e.g., … et al.) of the authors.

  12. line 189, “each of the circle images containing about 600 pixels points in total”, it does not make sense that a radar image only contains 600 pixels.

  13. line 208, 600-2100 m is the physical distance, not the pixel number unless the pixel spacing in your image is 1 m. The range values here should be the pixel number.

  14. Fig.1, the latitude and longitude information are missing and should be added to the x axis and y axis of the figure. Also, the figure looks like a screenshot and the resolution is low. Also, the font size of some sentences inside the figure is too small.

  15. line 219, you should state what method you use for rain removal. Also, is 1493 the number of images or image sequences? If it is the image sequence, what is the time length of each sequence? How many images are contained in each sequence?

  16. An example figure should be presented to show the radar image before and after rain removal.

  17. Fig. 2, the figure is a screenshot with low resolution and has weird black lines as boundary. The original image should be presented instead of the screenshot. The font size is also too small in some places. A colorbar should be included for the rgb image. The space between the figure and the sentences above should be increased. These problems also happen to all the figures below and should be corrected.

  18. line 270, add “the calculation of” before “fast-convergence gray-level”. 

  19. Although it is good to have a flowchart of the method like Fig.4, it is highly recommended to add figures showing intermediate results in each step shown in Fig. 4 so that the methodology and corresponding results look more convincing.

  20. Compared to previous methods for wind direction estimation (e.g., the Cosine fitting by Lund et al.), the method is pretty complicated with lots of calculations, which makes me wonder what’s the calculation efficiency of the method compared to the method in Lund et al. (ref [15])? Can the method be used for real time wind estimation?

  21. X-band marine radar images should be in grayscale with one channel. However, RGB images are presented in manuscript and it is unclear how they are converted from the one-channel image? 

  22. line 278, the variable expression at the beginning looks wrong. What does the square shape mean?

  23. line 282, how are the three alpha parameters determined?

  24. line 356, same problem as line 278.

  25. line 641, what does the parameter “k” mean? The variable “k” appears multiple times in the manuscript. 

  26. Table 2, what are the units of KLD, RMSE, and PCC?

  27. Estimation results presented in Table 2 are not consistent with results presented in Figs. 8 and 9. For example, when K is 16 the RMSE is around 2.2. However, as shown in Fig. 8, the average difference between reference wind direction and estimation when K=16 is around 100 degrees.

  28. It is weird that all estimation follows the exact same changing trend as the reference wind data without any fluctuations, which makes me wonder if there is something wrong with the experiment. For example, when rain is present, the estimation from radar should be affected.

  29. line 795, same problem as issue 1 above.

Author Response

(The authors gave the same response as above.)

Round 2

Reviewer 1 Report

 Accept after minor revision (corrections to minor methodological errors and text editing)

Author Response

The authors gratitude the help from reviewer, and will carefully handle the issues and errors in the paper. 

Reviewer 2 Report

The autors claimed that they have figures with improved quality but in contrast, the figure quality in the revised paper looks even worse. Please resubmit everything with improved image quality and double check the pdf file before submission.

Author Response

The author gratitude the help from reviewer, and will provide original figures to editor if necessary to avoid display blur issue, and the new PDF is modifyed for all blur figures. 
